# Learning and using language via recursive pragmatic reasoning about other agents

**Nathaniel J. Smith**[*]
University of Edinburgh

**Noah D. Goodman**
Stanford University

**Michael C. Frank**
Stanford University

## Abstract

Language users are remarkably good at making inferences about speakers' intentions in context, and children learning their native language also display substantial skill in acquiring the meanings of unknown words. These two cases are deeply related: Language users invent new terms in conversation, and language learners learn the literal meanings of words based on their pragmatic inferences about how those words are used. While pragmatic inference and word learning have both been independently characterized in probabilistic terms, no current work unifies these two. We describe a model in which language learners assume that they jointly approximate a shared, external lexicon and reason recursively about the goals of others in using this lexicon. This model captures phenomena in word learning and pragmatic inference; it additionally leads to insights about the emergence of communicative systems in conversation and the mechanisms by which pragmatic inferences become incorporated into word meanings.

## 1   Introduction

Two puzzles present themselves to language users: What do words mean in general, and what do they mean in context? Consider the utterances "it's raining," "I ate some of the cookies," or "can you close the window?" In each, a listener must go beyond the literal meaning of the words to fill in contextual details ("it's raining here and now"), infer that a stronger alternative is not true ("I ate some but not all of the cookies"), or more generally infer the speaker's communicative goal ("I want you to close the window right now because I'm cold"), a process known as *pragmatic reasoning*. Theories of pragmatics frame the process of language comprehension as inference about the generating goal of an utterance given a rational speaker [14, 8, 9]. For example, a listener might reason, "if she had wanted me to think 'all' of the cookies, she would have said 'all'—but she didn't. Hence 'all' must not be true and she must have eaten some *but not all* of the cookies." This kind of reasoning is core to language use.

But pragmatic reasoning about meaning-in-context relies on stable literal meanings that must themselves be learned. In both adults and children, uncertainty about word meanings is common, and often considering speakers' pragmatic goals can help to resolve this uncertainty. For example, if a novel word is used in a context containing both a novel and a familiar object, young children can make the inference that the novel word refers to the novel object [22].[1] For adults who are proficient language users, there are also a variety of intriguing cases in which listeners seem to create situation- and task-specific ways of referring to particular objects. For example, when asked to refer to idiosyncratic geometric shapes, over the course of an experimental session, participants create conventionalized descriptions that allow them to perform accurately even though they do not begin with shared labels [19, 7]. In both of these examples, reasoning about another person's goals informs

---

[*]nathaniel.smith@ed.ac.uk

[1]Very young children make inferences that are often labeled as "pragmatic" in that they involve reasoning about context [6, 1], though in some cases they are systematically 'too literal' (e.g. failing to strengthen SOME to SOME-BUT-NOT-ALL [23]). Here we remain agnostic about the age at which children are able to make such inferences robustly, as it may vary depending on the linguistic materials being used in the inference [2].

language learners' estimates of what words are likely to mean.

Despite this intersection, there is relatively little work that takes pragmatic reasoning into account when considering language learning in context. Recent work on grounded language learning has attempted to learn large sets of (sometimes relatively complex) word meanings from noisy and ambiguous input (e.g. [10, 17, 20]). And a number of models have begun to formalize the consequences of pragmatic reasoning in situations where limited learning takes place [12, 9, 3, 13]. But as yet these two strands of research have not been brought together so that the implications of pragmatics for learning can be investigated directly.

The goal of our current work is to investigate the possibilities for integrating models of recursive pragmatic reasoning with models of language learning, with the hope of capturing phenomena in both domains. We begin by describing a proposal for bringing the two together, noting several issues in previous approaches based on recursive reasoning under uncertainty. We next simulate findings on pragmatic inference in one-shot games (replicating previous work). We then build on these results to simulate the results of pragmatic learning in the language acquisition setting where one communicator is uncertain about the lexicon and in iterated communication games where both communicators are uncertain about the lexicon.

## 2 Model

We model a standard communication game [19, 7]: two participants each, separately, view identical arrays of objects. On the *Speaker's* screen, one object is highlighted; their goal is to get the *Listener* to click on this item. To do this, they have available a fixed, finite set of words; they must pick one. The Listener then receives this word, and attempts to guess which object the Speaker meant by it. In the psychology literature, as in real-world interactions, games are typically iterated; one view of our contribution here is as a generalization of one-shot models [9, 3] to the iterated context.

**2.1 Paradoxes in optimal models of pragmatic learning.** Multi-agent interactions are difficult to model in a normative or optimal framework without falling prey to paradox. Consider a simple model of the agents in the above game. First we define a *literal listener* $L_0$. This agent has a *lexicon* of associations between words and meanings; specifically, it assigns each word $w$ a vector of numbers in $(0, 1)$ describing the extent to which this word provides evidence for each possible object[2]. To interpret a word, the literal listener simply re-weights their prior expectation about what is referred to using their lexicon's entry for this word:

$$P_{L_0}(\text{object}|\text{word}, \text{lexicon}) \propto \text{lexicon}(\text{word}, \text{object}) \times P_{\text{prior}}(\text{object}). \tag{1}$$

Because of the normalization in this equation, there is a systematic but unimportant symmetry among lexicons; we remove this by assuming the lexicon sums to 1 over objects for each word. Confronted with such a listener, a speaker who chooses approximately optimal actions should attempt to choose a word which soft-maximizes the probability that the listener will assign to the target object—modulated by the effort or cost associated with producing this word:

$$P_{S_1}(\text{word}|\text{object}, \text{lexicon}) \propto \exp\left(\lambda\big(\log P_{L_0}(\text{object}|\text{word}, \text{lexicon}) - \text{cost}(\text{word})\big)\right). \tag{2}$$

But given this speaker, then the naive $L_0$ strategy is not optimal. Instead, listeners should use Bayes rule to invert the speaker's decision procedure [9]:

$$P_{L_2}(\text{object}|\text{word}, \text{lexicon}) \propto P_{S_1}(\text{word}|\text{object}, \text{lexicon}) \times P_{\text{prior}}(\text{object}). \tag{3}$$

Now a difficulty becomes apparent. Given such a listener, it is no longer optimal for speakers to implement strategy $S_1$; instead, they should implement strategy $S_3$ which soft-maximizes $P_{L_2}$ instead of $P_{L_0}$. And then listeners ought to implement $L_4$, and so on.

One option is to continue iterating such strategies until reaching a fixed point equilibrium. While this strategy guarantees that each agent will behave normatively given the other agent's strategy, there is no guarantee that such strategies will be near the system's global optimum. More importantly,

there is a great deal of evidence that humans do not use such equilibrium strategies; their behavior in language games (and in other games [5]) can be well-modeled as implementing $S_k$ or $L_k$ for some small $k$ [9]. Following this work, we recurse a finite (small) number of times, $n$. The consequence is that one agent, implementing $S_n$, is fully optimal with respect to the game, while the other, implementing $L_{n-1}$, is only nearly optimal—off by a single recursion.

This resolves one problem, but as soon as we attempt to add uncertainty about the meanings of words to such a model, a new paradox arises. Suppose the listener is a young child who is uncertain about the lexicon their partner is using. The obvious solution is for them to place a prior on the lexicon; they then update their posterior based on whatever utterances and contextual cues they observe, and in the mean time interpret each utterance by making their best guess, marginalizing out this uncertainty. This basic structure is captured in previous models of Bayesian word learning [10]. But when combined with the recursive pragmatic model, a new question arises: Given such a listener, what model should the speaker use? A rational speaker attempts to maximize the listener's likelihood of understanding, so if an uncertain listener interpets by marginalizing over some posterior, then a fully knowledgeable speaker should disregard their own lexical knowledge, and instead model and marginalize over the listener's uncertainty. But if they do this, then their utterances will provide no data about their lexicon, and there is nothing for the rational listener to learn from observing them.[3]

One final problem is that under this model, when agents switch roles between listener and speaker, there is nothing constraining them to continue using the same language. Optimizing task performance requires my lexicon as a speaker to match your lexicon as a listener and vice-versa, but there is nothing that relates my lexicon as a speaker to my lexicon as a listener, because these never interact. This clearly represents a dramatic mismatch to typical human communication, which almost never proceeds with distinct languages spoken by each participant.

**2.2   A conventionality-based model of pragmatic word learning.** We resolve the problems described above by assuming that speakers and listeners deviate from normative behavior by assuming a conventional lexicon. Specifically, our final convention-based agents assume: (a) There is some single, specific literal lexicon which everyone should be using, (b) and everyone else knows this lexicon, and believes that I know it as well, (c) but in fact I don't. These assumptions instantiate a kind of "social anxiety" in which agents are all trying to learn the correct lexicon that they assume everyone else knows.

Assumption (a) corresponds to the lexicographer's illusion: Naive language users will argue vociferously that words have specific meanings, even though these meanings are unobservable to everyone who purportedly uses them. It also explains why learners speak the language they hear (rather than some private language that they assume listeners will eventually learn): Under assumption (a), observing other speakers' behavior provides data about not just that speaker's idiosyncratic lexicon, but the consensus lexicon. Assumption (b) avoids the explosion of hyper$^n$-distributions described above: If agent $n$ knows the lexicon, they assume that all lower agents do as well, reducing to the original tractable model without uncertainty. And assumption (c) introduces a limited form of uncertainty at the top level, and thus the potential for learning. To the extent that a child's interlocutors do use a stable lexicon and do not fully adapt their speech to accomodate the child's limitations, these assumptions make a reasonable approximation for the child language learning case. In general, though, in arbitrary multi-turn interactions in which both agents have non-trivial uncertainty, these assumptions are incorrect, and thus induce complex and non-normative learning dynamics.

Formally, let an unadorned $L$ and $S$ denote the listener and speaker who follow the above assumptions. If the lexicon were known then the listener would draw inferences as in $L_{n-1}$ above; but by assumption (c), they have uncertainty, which they marginalize out:

$$P_L(\text{object}|\text{word}, L\text{'s data}) = \int P_{L_{n-1}}(\text{object}|\text{word}, \text{lexicon})P(\text{lexicon}|L\text{'s data})\, d(\text{lexicon}) \quad (4)$$

| Phenomenon | Ref. | WL | PI | PI+U | PI+WL | Section |
|---|---|---|---|---|---|---|
| Interpreting scalar implicature | [14] | | x | x | x | 3.1 |
| Interpreting Horn implicature | [15] | | | x | x | 3.2 |
| Learning literal meanings despite scalar implicature | [21] | | | | x | 4.1 |
| Disambiguating new words using old words | [22] | x | | x | x | 4.2 |
| Learning new words using old words | [22] | x | | | x | 4.2 |
| Disambiguation without learning | [16] | | | x | x | 4.2 |
| Emergence of novel & efficient lexicons | [11] | | | | x | 5.1 |
| Lexicalization of Horn implicature | [15] | | | | x | 5.2 |

Table 1: Empirical results and references. WL refers to the word learning model of [10]; PI refers to the recursive pragmatic inference model of [9]; PI+U refers to the pragmatic inference model of [3] which includes lexical uncertainty, marginalizes it out, and then recurses. Our current model is referred to here as PI+WL, and combines pragmatic inference with word learning.

Here $L$'s data consists of her previous experience with language. In particular in the iterated games explored here it consists of $S$'s previous utterances together with whatever other information $L$ may have about their intended referents (e.g. from contextual clues). By assumption (b), $L$ treats these utterances as samples from the knowledgeable speaker $S_{n-2}$, not $S$, and thus as being informative about the lexicon. For instance, when the data is a set of fully observed word-referent pairs $\{w_i, o_i\}$:

$$P(\text{lexicon}|L\text{'s data}) \propto P(\text{lexicon}) \prod_i P_{S_{n-2}}(w_i|o_i, \text{lexicon}) \qquad (5)$$

The top-level speaker $S$ attempts to select the word which soft-maximizes their utility, with utility now being defined in terms of the informativity of the expectation (over lexicons) that the listener will have for the right referent[4]:

$$P_S(\text{word}|\text{object}, S\text{'s data}) \propto \qquad (6)$$
$$\exp\left(\lambda\left(\log \int P_{L_{n-1}}(\text{object}|\text{word}, \text{lexicon})P(\text{lexicon}|S\text{'s data})\, d(\text{lexicon}) - \text{cost}(\text{word})\right)\right)$$

Here $P(\text{lexicon}|S\text{'s data})$ is defined similarly, when $S$ observes $L$'s interpretations of various utterances, and treats them as samples from $L_{n-1}$, not $L$. However, notice that if $S$ and $L$ have the same subjective distributions over lexicons, then $S$ is approximately optimal with respect to $L$ in the same sense that $S_k$ is approximately optimal with respect to $L_{k-1}$. In one-shot games, this model is conceptually equivalent to that of [3] restricted to $n = 3$; our key innovations are that we allow learning by replacing their $P(\text{lexicon})$ with $P(\text{lexicon}|\text{data})$, and provide a theoretical justification for how this learning can occur.

In the remainder of the paper, we apply the model described above to a set of one-shot pragmatic inference games that have been well-studied in linguistics [14, 15] and are addressed by previous one-shot models of pragmatic inference [9, 3]. These situations set the stage for simulations investigating how learning proceeds in iterated versions of such games, described in the following section. Results captured by our model and previous models are summarized in Table 1. In our simulations throughout, we somewhat arbitrarily set the recursion depth $n = 3$ (the minimal value that produces all the qualitative phenomena), $\lambda = 3$, and assume that all agents have shared priors on the lexicon and full knowledge of the cost function. Inference is via importance sampling from a Dirichlet prior over lexicons.

## 3 Pragmatic inference in one-shot games

**3.1 Scalar implicature.** Many sets of words in natural language form scales in which each term makes a successively stronger claim. "Some" and "all" form a scale of this type. While "I ate some

of the cookies" is compatible with the followup "in fact, I ate *all* of the cookies," the reverse is not true. "Might" and "must" are another example, as are "OK," "good," and "excellent." All of these scales allow for *scalar implicatures* [14]: the use of a less specific term pragmatically implies that the more specific term does not apply. So although "I ate some of the cookies" could in principle be compatible with eating ALL of them, the listener is lead to believe that SOME-BUT-NOT-ALL is the likely state of affairs. The recursive pragmatic reasoning portions of our model capture findings on scalar implicature in the same manner as previous models [3, 13].

**3.2    Horn implicature.** Consider a world which contains two words and two types of objects. One word is expensive to use, and one is cheap (call them "expensive" and "cheap" for short). One object type is common and one is rare; denote these COMMON and RARE. Intuitively, there are two possible communicative systems here: a good system where "cheap" referes to COMMON and "expensive" refers to RARE, and a bad system where the opposite holds. Obviously we would prefer to use the good system, but it has historically proven very difficult to derive this conclusion in a game theoretic setting, because both systems are stable equilibria: if our partner uses the bad system, then we would rather follow and communicate at some cost than switch to the good system and fail entirely [3].

Humans, however, unlike traditional game theoretic models, do make the inference that given two otherwise equivalent utterances, the costly utterance should have a rare or unusual meaning. We call this pattern *Horn implicature*, after [15]. For instance, "Lee got the car to stop" implies that Lee used an unusual method (e.g. not the brakes) because, had he used the brakes, the speaker would have chosen the simpler and shorter (less costly) expression, "Lee stopped the car" [15]. Surprisingly, Bergen et al. [3] show that the key to achieving this favorable result is ignorance. If a listener assigns equal probability to her partner using the good system or the bad system, then their best bet is to estimate $P_S(\text{word}|\text{object})$ as the average of $P_S(\text{word}|\text{object}, \text{good system})$ and $P_S(\text{word}|\text{object}, \text{bad system})$. These might seem to cancel out, but in fact they do not. In the good system, the utilities of the speaker's actions are relatively strongly separated compared to the bad system; therefore, a soft-max agent in the bad system has noiser behavior than in the good system, and the behavior in the good system dominates the average. Similar reasoning applies to an uncertain speaker. For example, in our model with a uniform prior over lexicons and $P_{\text{prior}}(\text{COMMON}) = 0.8$, cost("cheap") $= 0.5$, cost("expensive") $= 1.0$, the symmetry breaks in the appropriate way: Despite total ignorance about the conventional system, our modeled speakers prefer to use simple words for common referents ($P_S(\text{"cheap"}|\text{COMMON}) = 0.88$, $P_S(\text{"cheap"}|\text{RARE}) = 0.46$), and listeners show a similar bias ($P_L(\text{COMMON}|\text{"cheap"}) = 0.77$, $P_L(\text{COMMON}|\text{"expensive"}) = 0.65$).

This preference is weak; the critical point is that it exists at all, given the unbiased priors. We return to this in §5.2. [3] report a much stronger preference, which they accomplish by applying further layers of pragmatic recursion on top of these marginal distributions. On the one hand, this allows them to better fit their empirical data; on the other, it removes the possibility of learning the literal lexicon that underlies pragmatic inference – further recursion above the uncertainty means that it is only hypothetical agents who are ignorant, while the actual speaker and listener have no uncertainty about each other's generative process.

## 4    Pragmatics in learning from a knowledgable speaker

**4.1    Learning literal meanings despite scalar implicatures.** The acquisition of quantifiers like "some" provides a puzzle for most models of word learning: given that in many contexts, the word "some" is used to mean SOME-BUT-NOT-ALL, how do children learn that SOME-BUT-NOT-ALL is not in fact its literal meaning? Our model is able to take scalar implicatures into account when learning, and thus provide a potential solution, congruent with the observation that no known language in fact lexicalizes SOME-BUT-NOT-ALL [21].

Following the details of §3.1, we created a simulation in which the model's prior fixed the meaning of "all" to be a particular set ALL, but was ambiguous about whether "some" literally meant SOME-BUT-NOT-ALL (incorrect) or SOME-BUT-NOT-ALL OR ALL (correct). The model was then exposed to training situations in which "some" was used to refer to SOME-BUT-NOT-ALL. Despite this training, the model maintained substantial posterior probability on the correct hypothesis about the meaning of "some." Essentially, the model reasoned that although it had unambiguous evidence for "some" being used to refer to SOME-BUT-NOT-ALL, this was nonetheless consistent with a literal meaning of SOME-BUT-NOT-ALL OR ALL which had then been pragmatically strengthened.

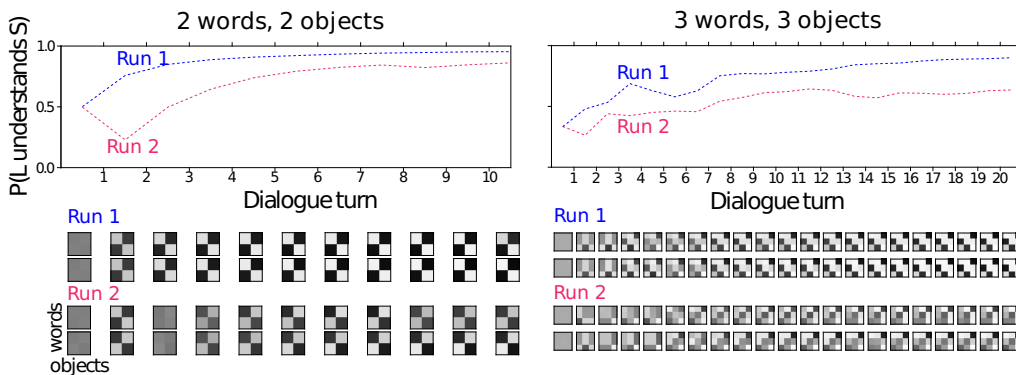

Figure 1: Simulations of two pragmatic agents playing a naming game. Each panel shows two representative simulation runs, with run 1 chosen to show strong convergence and run 2 chosen to show relatively weaker convergence. At each stage, $S$ and $L$ have different, possibly contradictory posteriors over the conventional, consensus lexicon. From these posteriors we derive the probability $P(L$ understands $S)$ (marginalizing over target objects and word choices), and also depict graphically $S$'s model of the listener (top row), and $L$'s actual model (bottom row).

Thus, a pragmatically-informed learner might be able to maintain the true meaning of SOME despite seemingly conflicting evidence.

**4.2 Disambiguation using known words.** Children, when presented with both a novel and a familiar object (e.g. an eggbeater and a ball), will treat a novel label (e.g. "dax") as referring to the novel object, for example by supplying the eggbeater when asked to "give me the dax" [22]. This phenomenon is sometimes referred to as "mutual exclusivity." Simple probabilistic word learning models can produce a similar pattern of findings [10], but all such models assume that learners retain the mapping between novel word and novel object demonstrated in the experimental situation. This observation is contradicted, however, by evidence that children often do not retain the mappings that are demonstrated by their inferences in the moment [16].

Our model provides an intriguing possible explanation of this finding: when simulating a single disambiguation situation, the model gives a substantial probability (e.g. 75%) that the speaker is referring to the novel object. Nevertheless, this inference is not accompanied by an increased belief that the novel word literally refers to this object. The learner's interpretation arises not from lexical mapping but instead from a variant of scalar implicature: the listener knows that the familiar word *does not* refer to the novel object—hence the novel word will be the best way to refer to the novel object, even if it literally could refer to either. Nevertheless, on repeated exposure to the same novel word, novel object situation, the learner does learn the mapping as part of the lexicon (congruent with other data on repeated training on disambiguation situations [4]).

## 5 Pragmatic reasoning in the absence of conventional meanings

**5.1 Emergence of efficient communicative conventions.** Experimental results suggest that communicators who start without a usable communication system are able to establish novel, consensus-based systems. For example, adults playing a communication game using only novel symbols with no conventional meaning will typically converge on a set of new conventions which allow them to accomplish their task [11]. Or in a less extreme example, communicators asked to refer to novel objects invent conventional names for them over the course of repeated interactions (e.g., "the ice skater" for an abstract figure vaguely resembling an ice skater, [7]). From a pure learning perspective this behavior is anomalous, however: Since both agents know perfectly well that there is no existing convention to discover, there is nothing to learn from the other's behavior. Furthermore, even if only one partner is producing the novel expressions, their behavior in these studies still becomes more regular (conventional) over time, which would seem to rule out a role for learning—even if there is some pattern in the expressions the speaker chooses to use, there is certainly nothing for the *speaker* to learn by observing these patterns, and thus their behavior should not change over time.

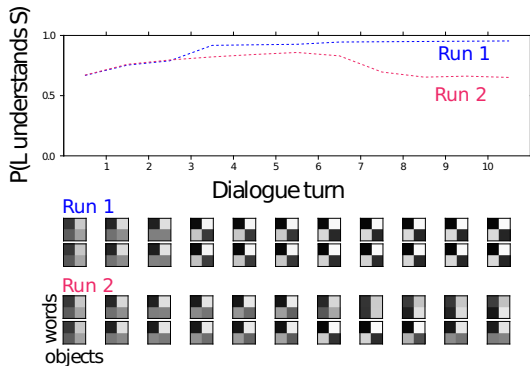

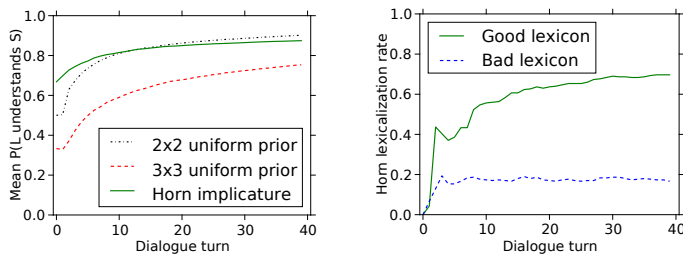

Figure 2: Example simulations showing the lexicalization of Horn implicatures. Plotting conventions are as above. In the first run, speaker and listener converge on a sparse and efficient communicative equilibrium, in which "cheap" means COMMON and "expensive" means RARE, while in the second they reach a sub-optimal equilibrium. As shown in Fig. 3, the former is more typical.

Figure 3: Averaged behavior over 300 dialogues as in Figs. 1 and 2. Left: Communicative success by game type and dialogue turn. Right: Proportion of dyads in the Horn implicature game (§5.2) who have converged on the 'good' or 'bad' lexicons and believe that these are literal meanings.

To model such phenomena, we imagine two agents playing the simple referential game introduced in § 2. On each turn the speaker is assigned a target object, utters some word referring to this object, the listener makes a guess at the object, and then, critically, the speaker observes the listener's guess and the listener receives feedback indicating the correct answer (i.e., the speaker's intended referent). Both agents then update their posterior over lexicons before proceeding to the next trial. As in [19, 7], the speaker and listener remain fixed in the same role throughout.

Fig. 1 shows the result of simulating several such games when both parties begin with a uniform prior over lexicons. Notice that: (a) agents' performance begins at chance, but quickly rises – a communicative system emerges where none previously existed; (b) they tend towards structured, sparse lexicons with a one-to-one correspondence between objects and words – these communicative systems are biased towards being useful and efficient; and (c) as the speaker and listener have entirely different data (the listener's interpretations and the speaker's intended referent, respectively), unlucky early guesses can lead them to believe in entirely contradictory lexicons—but they generally recover and converge. Each agent effectively uses their partner's behavior as a basis for forming weak beliefs about the underlying lexicon that they assume must exist. Since they then each act on these beliefs, and their partner uses the resulting actions to form new beliefs, they soon converge on using similar lexicons, and what started as a "superstition" becomes normatively correct. And unlike some previous models of emergence across multiple generations of agents [18, 25], this occurs within individual agents in a single dialogue.

**5.2  Lexicalization and loss of Horn implicatures.** A stronger example of how pragmatics can create biases in emerging lexicons can be observed by considering a version of this game played in the "cheap"/"expensive"/COMMON/RARE domain introduced in our discussion of Horn implicature (§3.2). Here, a uniform prior over lexicons, combined with pragmatic reasoning, causes each agent to start out weakly biased towards the associations "cheap" $\leftrightarrow$ COMMON, "expensive" $\leftrightarrow$ RARE. A fully rational listener who observed an uncertain speaker using words in this manner would therefore discount it as arising from this bias, and conclude that the speaker was, in fact, highly uncertain. Our convention-based listener, however, believes that speakers do know which convention is in use, and therefore tends to misinterpret this biased behavior as positive evidence that the 'good' system is in use. Similarly, convention-based speakers will wager that since on average they will succeed more often if listeners are using the 'good' system, they might as well try it. When they succeed, they take their success as evidence that the listener was in fact using the good system all along. As a result, dyads in this game end up converging onto a stable system at a rate far above chance, and

preferentially onto the 'good' system (Figs. 2 and 3).

In the process, though, something interesting happens. In this model, Horn implicatures depend on uncertainty about literal meaning. As the agents gather more data, their uncertainty is reduced, and thus through the course of a dialogue, the implicature is replaced by a belief that "cheap" *literally* means COMMON (and did all along). To demonstrate this phenomenon, we queried each agent in each simulated dyad about how they would refer to or interpret each object and word, *if* the two objects were equally common, which cancels the Horn implicature. As shown in Fig. 3 (right), after 30 turns, in nearly 70% of dyads both $S$ and $L$ used the 'good' mapping even in this implicature-free case, while less than 20% used the 'bad' mapping (with the rest being inconsistent).

This points to a fundamental difference in how learning interacts with Horn versus scalar implicatures. Depending on the details of the input, it is possible for our convention-based agents to observe pragmatically strengthened uses of scalar terms (e.g., "some" used to refer to SOME-BUT-NOT-ALL), without becoming confused into thinking that "some" *literally* means SOME-BUT-NOT-ALL (§4.1). This occurs because scalar implicature depends only on recursive pragmatic reasoning (§2.1), which our convention-based agents' learning rules are able to model and correct for. But, while our agents are able to use Horn implicatures in their own behaviour (§ 3.2), this happens implicitly as a result of their uncertainty, and our agents do not model the uncertainty of other agents; thus, when they observe other agents using Horn implicatures, they cannot interpret this behavior as arising from an implicature. Instead, they take it as reflecting the actual literal meaning. And this result isn't just a technical limitation of our implementation, but is intrinsic to our convention-based approach to combining pragmatics and learning: in our system, the only thing that makes word learning possible at all is each agent's assumption that other agents are better informed; otherwise, other agents' behavior would not provide any useful data for learning. Our model therefore makes the interesting prediction that all else being equal, uncertainty-based implicatures should over time be more prone to lexicalizing and becoming part of literal meaning than recursion-based implicatures are.

# 6   Conclusion

Language learners and language users must consider word meanings both within and across contexts. A critical part of this process is reasoning pragmatically about agents' goals in individual situations. In the current work we treat agents communicating with one another as assuming that there is a shared conventional lexicon which they both rely on, but with differing degrees of knowledge. They then reason recursively about how this lexicon should be used to convey particular meanings in context. These assumptions allow us to create a model that unifies two previously separate strands of modeling work on language usage and acquisition and account for a variety of new phenomena. In particular, we consider new explanations of disambiguation in early word learning and the acquisition of quantifiers, and demonstrate that our model is capable of developing novel and efficient communicative systems through iterated learning within the context of a single simulated conversation.

Our assumptions produce a tractable model, but because they deviate from pure rationality, they must introduce biases, of which we identify two: a tendency for pragmatic speakers and listeners to accentuate useful, sparse patterns in their communicative systems (§5.1), and for short, 'low cost' expressions to be assigned to common objects (§5.2). Strikingly, both of these biases systematically drive the overall communicative system towards greater global efficiency. In the long term, these processes should leave their mark on the structure of the language itself, which may contribute to explaining how languages become optimized for effective communication [26, 24].

More generally, understanding the interaction between pragmatics and learning is a precondition to developing a unified understanding of human language. Our work here takes a first step towards joining disparate strands of research that have treated language acquisition and language use as distinct.

**Acknowledgments**

This work was supported in part by the European Commission through the EU Cognitive Systems Project Xperience (FP7-ICT-270273), the John S. McDonnell Foundation, and ONR grant N000141310287.

## Footnotes

[2]We assume words refer directly to objects, rather than to abstract semantic features. Our simplification is without loss of generality, however, because we can interpret our model as marginalizing over such a representation, with our literal $P_{\text{lexicon}}(\text{object}|\text{word}) = \sum_{\text{features}} P(\text{object}|\text{features}) P_{\text{lexicon}}(\text{features}|\text{word})$.

[3]Of course, in reality both parties will generally have some uncertainty, making the situation even worse. If we start from an uncertain listener with a prior over lexicons, then a first-level uncertain speaker needs a prior over priors on lexicons, a second-level uncertain listener needs a prior over priors over priors, etc. The original $L_0 \rightarrow S_1 \rightarrow \dots$ recursion was bad enough, but at least each step had a constant cost. This new recursion produces hyper$^n$-distributions for which inference almost immediately becomes intractable even in principle, since the dimensionality of the learning problem increases with each step. Yet, without this addition of new uncertainty at each level, the model would dissolve back into certainty as in the previous paragraph, making learning impossible.

[4]An alternative model would have the speaker take the expectation over informativity, instead of the informativity of the expectation, which would correspond to slightly different utility functions. We adopt the current formulation for consistency with [3].

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
