[Reviews · NeurIPS 2013]

Submitted by Assigned_Reviewer_3

This paper presents a probabilistic model for language learning. The authors cover the nature in which a pair of cooperative agents may work together to create an agreed-upon language.

One question I have is how this could possibly be implemented in real-world language learning situations. Your evaluation of the emergence of phenomenon seen in real world languages makes me think you are trying to model or learn something about what real world language evolution is like. In real-world language learning, what are the sets of lexicons one sums over? The number of lexicons must be very large (infinite?) in the real world. It seems like an Latent Dirichlet Allocation-style model would be better suited to this task, as new words and new objects are encountered over time.

In general this paper is well written but there were several places where the wording was confusing. At line 150 I'm unclear what the convention-based listener/speaker are. In section 5.2 it is unclear what the word "this" means on lines 375 & 376 - is it the horn implicature mapping of the uniform?

Table 1 could be clearer if it specified the sections in which each feature was covered. For example, I'm not sure where the disambiguation without mapping is covered. What does novel and efficient mean here?

Ironically, this paper contains a lot of linguistic jargon with no explicit definitions. Perhaps you are asking me to learn the technical language of linguistics by pragmatic reasoning? I'd argue that the general NIPS audience may not know many of these terms, and may get more out of this paper if some parts were made more clear. For example, it would be helpful to present simple and explicit definitions for specificity and horn implicature. The definitions for each of these are given in a round about way in sections 3.1 and 3.2, but one of the cited papers (Bergen, Goodman, Levy) does it much more succinctly "specificity implicatures (less specific utterances imply the negation of more specific utterances) and Horn implicatures (more complex utterances are assigned to less likely meanings)" .

It's somewhat unfair to say that reference [11] doesn't produce efficient and novel languages. [11] explicitly explores efficiency. It would be more fair to have a row for efficient and a row for novel. It's not even clear to me what "novel" means in this paper, as all of the lexicons must be known beforehand. How can any of them be novel, then?

I appreciate the authors' thoughtful rebuttal, and have taken it into account.
Summary: This is an interesting approach to modeling how agents may work together to settle upon an agreed upon lexicon. This paper suffers from some linguistic jargon and clarity issues, but in general is an interesting exposition of pragmatic reasoning for language learning.

Submitted by Assigned_Reviewer_5

This paper puts forward a model of learning vocabulary, in the context of a hypothetical 'naming game'. A row of objects visible ton two players, the 'speaker' and the 'listener'. There are repeated rounds; in each round an object is indicated to the speaker, who must then say a word that will cause the listener to point to that object.

This game can be (and has been) set up to be played between people, but this paper considers simulated play between learning algorithms.

Both players start with prior beliefs about a lexicon; they mutually refine these beliefs during repeated play.

The paper starts with a nice discussion of different ways in which to model this vocabulary learning problem. In my view this is the best part of the paper. The authors point out that there can be deep recursions if both players attempt to optimise their communication strategy in a game theoretic manner, and they make an interesting point that this deep recursion does not seem to allow vocabulary learning. To prevent such recursion, the authors introduce what seems a nice idea that both players believe that there is some correct conventional lexicon, of which they themselves have incomplete knowledge. This allows apparently simpler shallow reasoning about what the lexicon may be.

There are then case studies of two learning problems: uninformative prior beliefs, but objects with different frequencies and words with different costs; and the case of specificity implicature, where the problem is to learn that 'some' is pragmatically used to mean some, though it can also mean all.

I feel that this is potentially a rather nice line of research, but I also feel that this paper has some problems in its current form.

First, there are rather few simulations; the results of a few individual simulations only are given, and there is no systematic empirical analysis under a range of conditions. How do these algorithms fare with slightly larger problems? I take the point that different amounts of experimentation with simulations are expected in different fields -- but the authors should realise that if they propose an algorithm, they should give sufficient simulations to give some empirical idea of how it behaves, and what its behaviour is on problems of different sizes. Traces of individual runs are not enough.

Second, I was a little unclear as to what the prior distributions over lexicons are. In the case of Horn implicature, each word is assumed to match a single object, so the prior can be represented as a multinomial distribution over the objects. In the case of specificity implicature, the word some can mean some or all, which seems to be represented as a probability distribution across objects: what is the prior distribution in this case? Thanks to the authors for clarifying this point.

Third, a rather simpler model would be for a learner to maintain a current assumed lexicon; if there are communication failures, then the learner may change it. The learner (and speaker?) may then repeatedly change their own lexicons, until they reach a state where the lexicons match, communication succeeds, and no further changes are necessary. This idea might turn out to need a Bayesian formulation -- but there could be simpler possibilities requiring less computation?

As a suggestion, what about two-word phrases? Can you produce a model of learning nouns when there are adjectives also? You could have arrays of objects with two features, and you have one word describing one feature, and the other word describing the other feature. You could only say 'red' if there is only one red object, but you could say 'red car' if there is a red car, a green car, and a red lorry. Your present examples are rather small...
Summary: The basic idea of this paper seems nice, but the research has perhaps not yet been developed far. The authors at least need to extend the experiments.

Learning for pragmatic communication seems a fascinating problem, and this seems a good initial attempt to produce a model for it.

This is potentially a high impact paper because they are trying to produce a new model for an interesting problem.

Submitted by Assigned_Reviewer_7

This paper brings a new idea to game-theoretic models of lexicon
acquisition: the assumption of a conventionalized lexicon. Each
speaker/listener believes that such a lexicon exists, that everyone
else is aware of it, and moreover, that the speaker/listener is
expected to be aware of it, even though s/he is not. This set of
assumptions greatly simplifies the complexity of prior game-theoretic
models of lexicon acquisition, resolving an "infinite recursion"
problem that seems to bear little relationship to real language use.
The model is shown to yield realistic predictions for several word
learning and pragmatic implicature phenomena. It is generally quite
well-written, although there are a few spelling mistakes.

My main suggestion for improvement would be to more closely relate the
theoretical predictions of the proposed model with empirical research
on language acquisition. Is there evidence that speakers really do
modify their speech based on their expectations about the listener's
lexical knowledge? And if so, is there evidence that they do this
optimally? I was also a little skeptical of the account of how
speakers acquire the literal meaning of words like "some";
intuitively, I would imagine that young children do start with the
definition "some-but-not-all", and only learn the literal definition
through explicit instruction; is there any empirical research on this?
Summary: This paper brings a new idea to game-theoretic models of lexicon acquisition: the assumption of a conventionalized lexicon. This idea has intuitive appeal, and is shown to capture a range of relevant phenomena.
Author Feedback

Author rebuttal: We see three main critiques raised by the reviewers:

1. Use of linguistic terminology was excessive or unclear.

These comments are extremely helpful for us in reaching the broad audience that we aim for in this work. If accepted we will be sure to take them into account when revising.

2. It would be more realistic to allow a richer model of literal meaning, allowing for multi-word phrases (Reviewer 5) and unbounded numbers of words and objects via non-parametric methods (Reviewer 3).

We agree completely with this comment. Both of these extensions are part of our larger research agenda (and unpublished work has investigated both). With respect to the current paper, however, we were attempting to strike a balance between clarity and sophistication. We see the heart of our work as the discussion of the contradictions between learning models and pragmatic reasoning models (sec. 2.1), and our proposal for resolving them (sec. 2.2). Right now the core literal learning part of our model (line 90) uses simple word/object associations, but in principle it would be straightforward to replace this component with any standard Bayesian language learning model (e.g. ref [25] or Xu & Tenenbaum 2007; see also our footnote 2). When integrated into the full model, though, even the current word/object associations end up producing rich and counter-intuitive behavior -- enough so that the bulk of the paper is taken up with explaining and demonstrating this behavior through simulations (sec. 3-5).

Using a more realistic model would add more complexity to be explained -- in fact, our current implementation does model multi-word utterances, but we disabled this for the current report because it didn't change any qualitative results. In addition, a more complex learning model would make it more difficult to isolate the effects of what we view as the primary contribution, the pragmatics/learning combination. For example, more realistic language learning models generally incorporate some sort of sparseness prior. We find that our model produces sparse lexicons (sec 5.1), and that it does so even when using our simple model which has no sparseness prior. This demonstrates that this bias is a (non-obvious) side-effect of the pragmatics/learning interplay. So we feel that establishing the behavior of this simple model is a necessary precondition to understanding or explaining the behavior of the more realistic models that will follow.

3. Limited number of simulations in the manuscript (Reviewer 5).

Again, we feel that there is a tradeoff between complexity and clarity in our current work. While it would be possible to run simulations with more words and objects, these simulations would go beyond the human experimental literature and might not provide insights into the behavior of the model with respect to the phenomena of interest. (We recognize that this may be a disciplinary issue, given that it is unusual in machine learning to see so much attention given to such small scale problems).

We would also like to clarify the simulation content of the current manuscript. Sections 3-5 present the results of six qualitatively different simulation studies. Figs. 1 & 2 present individual simulation runs for illustration, but population statistics for batch runs are given in Fig. 3. In future we plan to run human experiments to test some of the hypotheses generated by these simulations (e.g., the explanation of disambiguation without mapping given in lines 292-305, and the discrepancy between Horn and specificity implicatures with regard to iterated learning, lines 393-404). We also plan to expand the model to handle richer forms of linguistic structure. Both of these moves will provide new datasets for future work.

Again, we appreciate the thoughtful reviews. Responses to some specific queries:

Reviewer 3: What we mean by a "novel" lexicon is that our agents start jointly using lexicons that previously, no agent was using. What we mean by "efficient" is that this lexicon selection process is systematically biased towards choosing "good" ones (e.g. in the Horn implicature case, Fig. 3 right-hand side). Ref [11], by comparison, considers how to use an existing lexicon efficiently, which is a different thing. (Our model reduces to the model of [11] in the case where the lexicon is shared between communicators and known a priori.) Thank you for the suggestion about Table 1, we will revise accordingly.

Reviewer 5: In all cases, we represent the lexicon as a matrix of numbers between 0 and 1, with each row constrained to sum to 1. In the Horn implicature cases, our prior is simply a uniform distribution over such matrices. In the specificity implicature case, we use a prior which encodes the knowledge that the word "all" refers to the ALL object but not the SOME-BUT-NOT-ALL object (a Dirichlet with pseudocounts favoring ALL), and place a uniform prior on the meanings of "some." Thank you for the suggestion of a heuristic comparison model--we will investigate this in our future work.

Reviewer 7: There is substantial empirical evidence that speakers adjust what words they're using based on feedback from their listeners (e.g. ref. [8]). On the other hand, it is unknown whether this behavior is optimal because ours is (to our knowledge) the first quantitative investigation of optimal behavior, so there hasn't been any way to check! Regarding the meaning of "some", there is a large body of research on children's quantifier use (reviewed in ref. [2]). To summarize: children do seem to have access to the literal meaning of "some" from an early stage. In fact it's not until age five or older that they are able to correctly infer that "some" implies SOME-BUT-NOT-ALL, and before this they seem, in the words of one auther, "more literal" than adults. So our simulations suggesting maintenance of the literal meaning (sec. 4.2) are at least prima facie consistent with the developmental literature.